# Biocompatible sulfonium-based covalent probes for endogenous tubulin fluorescence nanoscopy in live and fixed cells

Marie Auvray [1], Tanja Koenen[2], Olexandr Dybkov[3], Henning Urlaub [3,4] & Gražvydas Lukinavičius [1] ✉

Fluorescent probes enable precise visualization of dynamic cellular processes, especially when combined with super-resolution imaging techniques that overcome the diffraction limit. However, traditional labeling strategies, including fluorescent protein fusions (e.g., GFP) or ligand-linked fluorophores, often perturb protein function or induce biological side effects. Here, we report a covalent fluorescent probe for endogenous tubulin, a key cytoskeletal protein governing cell division, motility, and intracellular transport. Using cabazitaxel as a tubulin targeting moiety and silicon-rhodamine as a cell permeable fluorophore, we designed and optimized probe, **6-SiR-*o*-C₉-CTX**, containing a biocompatible cleavable linker with a sulfonium center. It exhibits cell permeability, fluorogenic behavior, and efficient covalent labeling of tubulin across multiple human cell lines. Importantly, taxane targeting moiety can be removed post-labeling, preserving tubulin's functions. This labeling strategy is compatible with STED nanoscopy in both live and fixed cells, enabling high-resolution, minimally invasive cytoskeletal imaging, and advancing the toolkit for studying dynamic cellular processes.

Fluorescence microscopy offers a possibility to visualize dynamic processes in living cells. In the recent decades, super-resolution microscopy has become an essential tool to understand these biological processes with an unrivaled precision by overcoming the diffraction limit[1]. These methods heavily relies on labeling techniques, which show a minimal influence on the function of the studied biomolecules[2,3]. In living cells, two strategies are mostly used for labeling of a protein-of-interest (POI). First, the fusion with a fluorescent protein (for example, the well-known Green Fluorescent Protein −GFP) or with a self-labeling tag that hosts a synthetic fluorophore[4]. Nevertheless, these strategies add a large extra domain to the POI: for example, GFP (27 kDa) has a length of 4.2 nm and a diameter of 2.4 nm[5]. This large size mainly causes two problems: mislocalization and function disruption of POI[6]. The second approach involves using

fluorescent probes, which consist of an organic fluorophore conjugated via a linker to a ligand that targets and non-covalently binds the POI. This method ensures the fluorescent tag is as small as possible. However, ligands can often exhibit biological effects and may occupy active sites on the protein, potentially altering its function. In this study, we aimed to explore an alternative labeling strategy for endogenous proteins that minimizes impact on their native function. In 2009, Hamachi introduced the ligand-directed labeling strategy[7]. This approach relies on proximity induced reaction between a cleavable linker and POI. Once the probe binds to the POI, the nucleophilic side chain of an amino acid nearby the binding site can react with the cleavable linker to create a covalent bond between the tag and the protein, while releasing the ligand. Over the last fifteen years, cleavable linkers that can react with various amino acids have been

---

[1]Chromatin Labeling and Imaging group, Department of NanoBiophotonics, Max Planck Institute for Multidisciplinary Sciences, Göttingen, Germany. [2]Department of NanoBiophotonics, Max Planck Institute for Multidisciplinary Sciences, Göttingen, Germany. [3]Bioanalytical Mass Spectrometry, Max Planck Institute for Multidisciplinary Sciences, Göttingen, Germany. [4]Bioanalytics Group, Institute for Clinical Chemistry, University Medical Center Göttingen, Göttingen, Germany. ✉e-mail: grazvydas.lukinavicius@mpinat.mpg.de

developed[8–12]. Despite the tremendous potential of this approach for fluorescence microscopy, it has not been extensively applied. In particular, its application in the field of fluorescence microscopy is mostly restricted to extracellular targets, like membrane receptors[13–16]. The main reason is that it is challenging to find a compromise between reactivity and stability of the cleavable linker inside cells. In addition, making these probes cell-permeable can be an issue.

In this study, we developed a series of covalent fluorescent probes for endogenous tubulin. This protein is highly abundant in mammalian cells, in which it represents 3–4% of the total proteins and up to 10% in brain[17]. Two subunits α- and β-tubulin form heterodimers and polymerize into microtubules, one of the main components of the cytoskeleton. Both proteins exist in numerous isotypes encoded by different genes. In human, there are 9 genes for α-tubulin and 10 for β-tubulin[18]. In addition, tubulin can undergo post-translational modifications, leading to a huge variety of forms in cells[19]. It is therefore involved in many different cellular processes like cell movement, division or trafficking of biomolecules. Previous studies have identified several tubulin-specific probes that are derived from well-established anticancer agents taxanes[20–22]. These probes bind tubulin reversibly and require continuous presence of the probes in the imaging medium to maintain optimal signal, which may cause cytotoxic effects during prolonged incubation. They are also unsuitable for cell fixation. To overcome these limitations, we present a covalent and biocompatible probe incorporating a cleavable linker with a sulfonium center. This design enables the separation of the targeting moiety (taxane) from the fluorescence signal-generating component during labeling, enhancing biocompatibility. Through in vitro and *in cellulo* evaluations, we identified silicon-rhodamine (SiR) probe **6-SiR-*o*-C₉-CTX** as an optimal fluorogenic, cell-membrane-permeable candidate. Importantly, we demonstrated that the ligand can be washed out while retaining robust tubulin staining, thus providing a labeling technique that minimally impacts protein structure and function. This probe enables covalent labeling of endogenous tubulin across various cell lines, making it suitable for stimulated emission depletion (STED) nanoscopy in both live and fixed cells.

## Results

### Design and synthesis of the probes

Tubulin-covalent probes are made of three parts: a dye, a ligand to direct the probe to the POI and a cleavable electrophile to create a covalent bond between the POI and the fluorophore, while releasing the ligand (Fig. 1a).

Many cleavable electrophiles have been developed in the past ten years, highlighting the importance of finding a compromise between the reactivity and the stability of the probe. We focused our attention on sulfonium, a biocompatible electrophile naturally found in cells across multiple species, where its derivatives play a crucial role in metabolism and cytoprotection[23–25]. In addition, recent proteomic studies by Z. Li demonstrated promising results for ligand-directed chemistry with *para*-benzyl sulfonium[26,27]. Rhodamine derivatives are among the most popular dyes to develop probes for live-cell imaging. They are in equilibrium between two forms: a nonfluorescent hydrophobic spirolactone (OFF state) and a fluorescent hydrophilic zwitterion (ON state) (Fig. 1a). In aqueous media, aggregation drives rhodamine to the spirolactone form, which is cell permeable. Once bound to its target, the interactions with the protein will push equilibrium towards the fluorescent zwitterionic form. One of the most prominent examples is SiR, which displays optimal fluorogenicity and cell permeability[28]. In our previous studies, we showed that the best performing ligand for tubulin in combination with SiR isomer-6 derivatives was cabazitaxel[20,29]. We thus performed a mini screening using silicon-rhodamine as dye and cabazitaxel (CTX) as ligand, and fine-tuned the structure by probing three different regioisomers (*ortho,*

*meta,* and *para*) of the cleavable electrophile (Fig. 1a and Supplementary Fig. 1).

The probes were synthesized via a convergent 7-step synthetic path with overall yields between 20 and 43% (Fig. 1b). First, thiols **7** were synthesized from the corresponding bromine derivatives, performing a nucleophilic substitution with thiourea followed by hydrolysis. Afterwards, thiols reacted with bromine derivatives of various chain lengths to give thioethers **8–13**. Sulfoniums with free amines and carboxylic acids **14–19** were obtained with yields ranging from 55 to 99% via a one-pot reaction in a mixture of formic and acetic acids in the presence of an excess of methyl trifluoromethanesulfonate. The synthesis ended with a four-step, one-pot reaction that links the cleavable electrophile to both the ligand and the dye. This last step allowed to get expected compounds with good yields ranging from 35 to 58%. Overall, this synthetic path is highly convergent, which allows a rapid access to structural diversity.

### Photophysical properties

Photophysical properties of the probes were investigated in phosphate-buffered saline (PBS), in PBS containing 0.5 mg/ml bovine serum albumin (BSA), in PBS containing 0.1% sodium dodecyl sulfate (SDS) or in presence of tubulin after 4 h at 37 °C to ensure complete tubulin polymerization (Fig. 1c, Supplementary Figs. 2, 3 and Supplementary Table 1). As mentioned previously, SiR-based probes are in equilibrium between two forms: a spirolactone and a zwitterionic forms. Multiple studies demonstrate that SiR equilibrium is highly sensitive to the environment[28,30,31]. In our recent study, DFT calculations of potential energies revealed that the spirolactone form is more stable in low-dielectric environments (e.g., dioxane), whereas the zwitterion form is preferred in high-dielectric environments (e.g., water)[32]. We have shown that in aqueous media (e.g., PBS), the hydrophobic SiR spirolactone tends to form aggregates which likely have a low internal dielectric constant[20]. This explains the low absorbance and emission observed in this media for most of the probes and recovery of absorbance and emission after SDS addition (Supplementary Figs. 2–4). However, probe **5**, containing a PEG linker, exhibits a fluorescence at least six times higher than other probes in aqueous media (Supplementary Fig. 3). This linker prevents aggregation by increasing water solubility, and therefore push the equilibrium towards the open fluorescent form. In the presence of SDS, aggregation is completely inhibited, with SiR adopting a zwitterionic form. This results in similar photophysical properties across all derivatives. Upon binding to tubulin, a shift in the spirocyclization equilibrium causes a 4- to 14-fold fluorescence enhancement relative to PBS for all probes except probe **5**. For probe **5**, only a modest increase in fluorescence (1.4-fold) is observed, due to its residual fluorescence in aqueous media. Among the probes, probe **2** demonstrates the highest fluorescence enhancement upon tubulin binding compared to PBS (14.3-fold, Supplementary Table 1). Notably, the non-target protein bovine serum albumin (BSA) induces only minimal changes in absorbance and fluorescence (Supplementary Fig. 3).

We assumed that the probes fluorescence in the presence of PBS containing 0.1% SDS corresponds to 100% of SiR fluorescent zwitterion. This allowed to estimate the equilibrium shift upon tubulin binding. We estimate that probes can reach up to 49% of fluorescent zwitterion content once bound to tubulin (Fig. 1c). Overall, probe **2** possesses the best photophysical properties, being the most fluorescent (49% of probe in ON state after binding to tubulin) and the most fluorogenic (14.3-fold fluorescence enhancement in the presence of tubulin compared to PBS).

### In vitro reaction with purified tubulin

Next, we checked the stability of the probes in aqueous media (PBS) at 37 °C as some probes containing cleavable linkers might be stable only for a few hours in solution[12]. Only minor degradation was

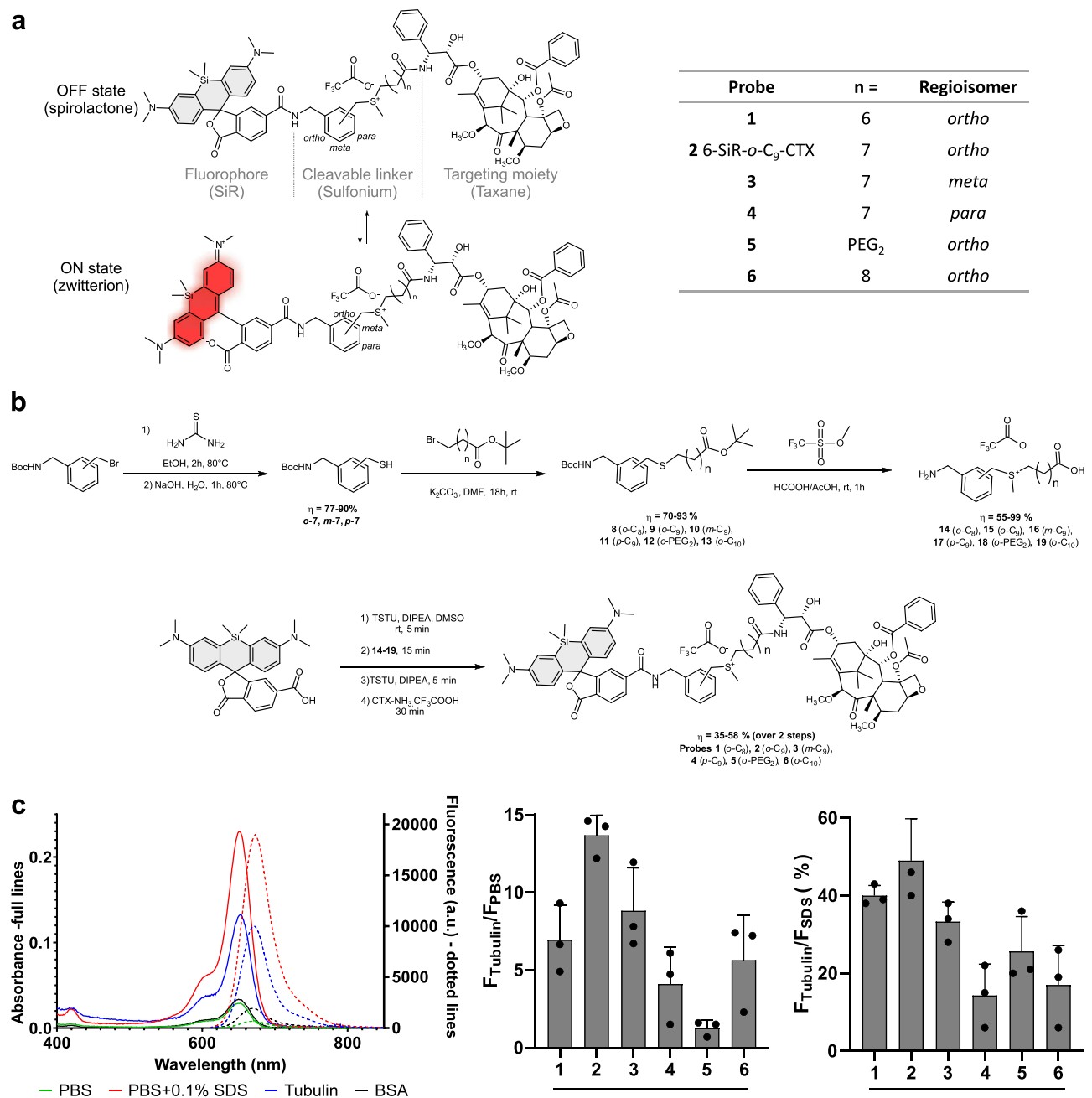

**Fig. 1 | Structure, synthesis and properties of the covalent probes. a** Structure of tubulin probes featuring a sulfonium-based cleavable linker and silicon-rhodamine (SiR) which can toggle between ON and OFF states. **b** Synthetic path of the covalent probes (**c**) Photophysical properties of the probes. Left: Absorption and emission spectra of **6-SiR-o-C₉-CTX** in presence of tubulin (blue), PBS (green), in presence of BSA (black) and PBS containing 0.1% SDS (red). Spectra are represented as averages of three independently repeated experiments (N = 3). Middle: Fluorescence increase upon tubulin binding, as compared to PBS. Right: Percent of dye open when bound to tubulin. Data are presented as mean of triplicate with standard deviation. Source data are provided as a Source data file.

observed, as more than 75% of the probes remain after 48 h at 37 °C (Supplementary Fig. 5). Major degradation pathways were hydrolysis (approx. 5–10% after 48 h) and intramolecular reaction (approx. 5% after 48 h), as the ligand contains some free nucleophilic groups. Prior to live-cell experiments, reactions between purified tubulin from porcine brain and the probes were conducted in vitro at 37 °C (Fig. 2, Supplementary Fig. 6). These probes are based on ligand-directed chemistry. They will first bind non-covalently tubulin, and thanks to proximity-induced reactivity, a nucleophilic residue nearby the binding pocket will react on the benzylic position next to the sulfonium center. It will therefore create a covalent bond between

the dye and the protein. The regioselectivity of the substitution on these sulfonium derivatives was previously established by et C. Dalhoff et al.[33]. Probes were used in 4-fold excess compared to tubulin in order to assess the selectivity of the reaction. The reaction was monitored by SDS–polyacrylamide gel electrophoresis (SDS-PAGE) and *in-gel* fluorescence analysis.

Apparent reaction rate was ranging from 0.12 to 0.28 h⁻¹ and it corresponds to a second order rate constant approx. 2–5 L mol⁻¹ s⁻¹, which is comparable to previously reported LDAI (ligand-directed acyl-imidazole chemistry) (Fig. 2c)[34]. On average, reactions with *ortho*-isomers were around 50% faster than with *meta* or *para*

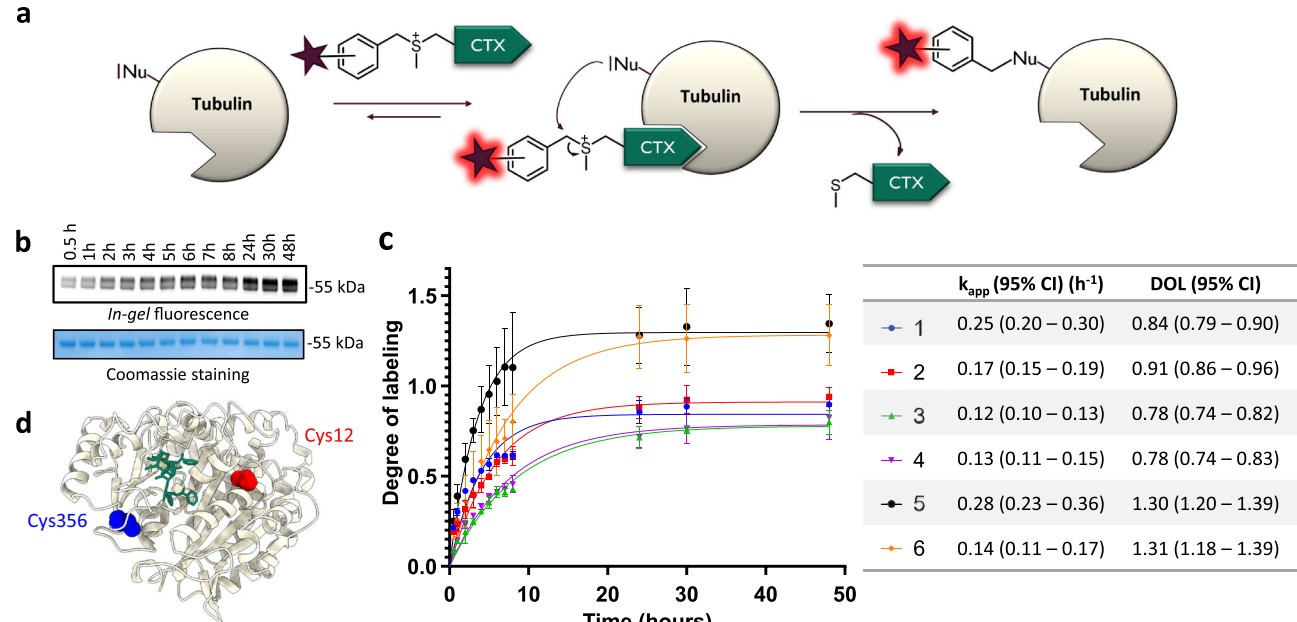

**Fig. 2 | In vitro labeling experiments of purified pig (Sus scrofa) tubulin (0.5 mg/mL ~5 μM) with the covalent probes (20 μM) over 48 h at 37 °C in General Tubulin Buffer. a** Reaction between the probes and tubulin. **b** Representative SDS-PAGE analysis of the labeling reaction. The gel was analyzed by *in-gel* fluorescence imaging (up) and stained with Coomassie Brilliant Blue (down). The experiments were performed in triplicate (N = 3) (**c**) Time-course of in vitro labeling of tubulin with the covalent probes. The experiments were performed in triplicate to obtain mean and standard deviation values (shown as error bars). Table contains mean values and 95% confidence intervals (CI) obtained from fitting data points to one phase association model from GraphPad Prism. **d** Structure of β-tubulin (from pig (sus scrofa) – PDB: 5SYF) with Taxol. The main labeling site is highlighted in blue (Cys 356), and the minor labeling site in red (Cys 12). Source data are provided as a Source data file.

isomers. Compounds can be divided into two groups based on the degree-of-labeling (DOL). Probes **1**–**4** reach DOL between 0.8 and 0.9, which corresponds to an optimal efficiency, close to the ideal DOL of 1. Decent labeling yield (around 50%) can be achieved within 2–3 h for Probe **1** and **6-SiR-*o*-C₉-CTX** (**2**). Probes **5** and **6** demonstrate DOL >1 suggesting multiple labeling sites and a lower selectivity (Fig. 2c). Despite the probe was put in excess (4 equivalents), the moderate DOL observed for all the probes is consistent with the proximity induced reactivity mechanism on which is based ligand-directed chemistry.

Afterwards, we also checked that the bond between the dye and tubulin was stable in aqueous conditions (Supplementary Fig. 7). Indeed, no degradation was observed after 48 h in PBS (pH = 7.4) at 37 °C. The labeling site(s) were then determined by proteolytic *in-gel* digestion followed by mass spectrometry. Seven different peptides were observed with a + 573 Da modification, corresponding to five different sequences labeled with SiR (Supplementary Figs. 8, 9, Supplementary Table 2 and Supplementary Data 1). For all these peptides, the major modified amino acid was a cysteine. This is in line with a previously reported study of cysteine labeling using sulfonium salts[26]. The high number of observed peptides can be explained by the numerous tubulin isotypes and isoforms contained in the sample. Indeed, most of the identified peptides come from different tubulin isotypes but correspond to the same modified amino acid. Overall, a primary labeling site and three secondary labeling sites were identified. The major labeling site was Cys356, located within the TAVCDIPPR/VAVCDIPPR peptide of β-tubulin (Supplementary Figs. 8, 9 and Supplementary Table 2). Additionally, a secondary labeling site was identified in β-tubulin at Cys12 within the peptide EIVHIQAGQCGNQIGAK, which MS1 precursor abundance was approximately ten times lower (Supplementary Fig. 8). Modeling demonstrates that both labeling sites are located near the taxane binding pocket inside lumen of microtubule, as illustrated in Fig. 2d, Supplementary Fig. 10 and Supplementary Movie 1.

Surprisingly, two minor labeling sites were also identified on α-tubulin (Cys347 and Cys376) despite cabazitaxel is known to bind β-tubulin. It might be explained by the close proximity of α and β subunit in microtubules. Nevertheless, the precursor abundance for the corresponding peptides was 20 times lower than for the major labeling site on β-tubulin, suggesting a really low labeling efficiency at these two positions on α-tubulin (Supplementary Fig. 8). The presence of several labeling sites might also explain the DOL above one obtained for Probe 5 and Probe 6.

Despite this experiment was done with porcine tubulin, there is high structural homology of tubulin among species[35]. The sequences of the different α and β-tubulin isotypes are highly conserved between human and pig as shown by sequence alignments and crystal structures (Supplementary Figs. 10–12). All the detected labeled peptides can also be found in human tubulin, including the main labeled peptide (AVCDIPPRGL), which is highly conserved. This suggests that analogous labeling might occur in other organisms and we can expect similar labeling sites using a human cell line for living cells experiments.

Overall, the in vitro tubulin labeling study highlighted the great potential of probes **1** and **2** for tubulin labeling in living cells. We have also shown that Cys356, which is nearby the taxane binding site on β-tubulin, is the main labeling site of the probe.

## Tubulin labeling in living cells

Probe cytotoxicity was first evaluated on HeLa CCL cells (Supplementary Fig. 13 and Supplementary Table 3). Taxane derivatives, such as cabazitaxel, stabilize microtubules and inhibit cell proliferation by arresting the cell cycle and preventing proper mitotic spindle assembly[36]. This disruption triggers apoptosis and DNA fragmentation. To assess the population of cells with fragmented DNA, corresponding to the SubG1 phase, we employed imaging cytometry. Because DNA fragmentation (appearance of the subG1

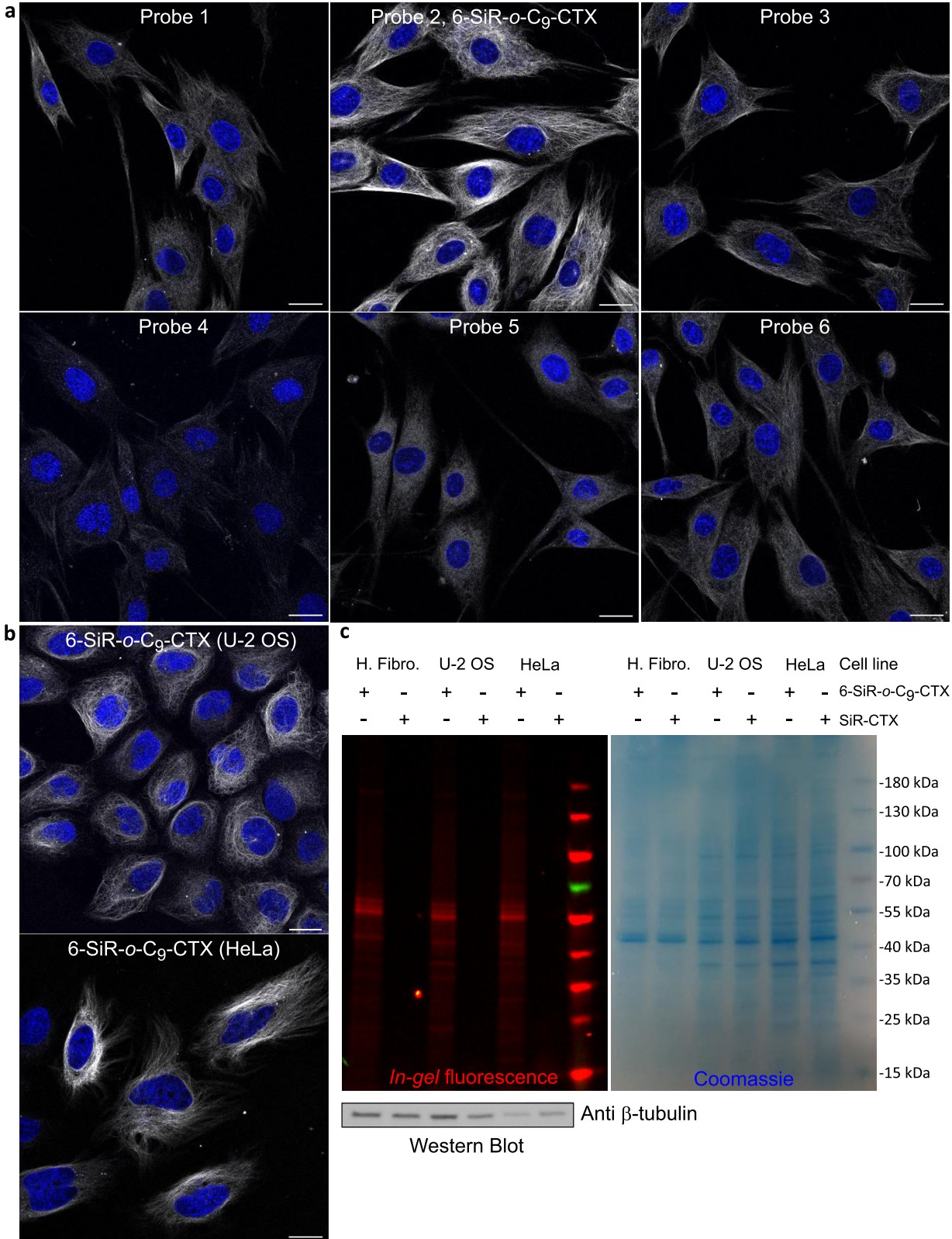

phase) can occur before the plasma membrane becomes permeable, standard cytotoxicity tests based on plasma membrane permeation are less sensitive than this cell cycle assay. After 24 h, most of the probes results in toxicity in the nanomolar range, with cytotoxicity thresholds ranging from 31.25 to 250 nM. These values are consistent with the toxicity of previously reported tubulin probe **SiR-CTX** (cytotoxicity threshold: 62.5 nM)[20]. We hypothesize that this measurement could be useful to assess and compare the cell permeability of the probes. According to the obtained values, *ortho* isomers are more permeable compared to *meta* and *para* isomers. Once more, probe **5** is an exception and seems to be less cell-permeable than other *ortho* isomers. PEG linker increases the solubility while reducing the cell permeability, by pushing the equilibrium towards the open form. It is consistent with the spectroscopic

**Fig. 3 | Tubulin labeling in living cells. a** Live human dermal fibroblasts stained with probes (1 µM in OptiMEM) and Hoechst 33342 nucleic acid dye (1 µg/mL) for 4 h. Confocal microscopy images were acquired using LEICA SP8. Gray channel ($\lambda_{ex}$ = 633 nm, $\lambda_{em}$ = 650–710 nm) corresponds to probe staining, and blue channel ($\lambda_{ex}$ = 405 nm, $\lambda_{em}$ = 415–480 nm) corresponds to Hoechst 33342 staining. Scale bar = 20 µm. This experiment was reproduced three times on different days with cells from different passages (**b**) Live-cell imaging of U-2 OS and HeLa CCL cells stained with **6-SiR-o-C₉-CTX** (1 µM in OptiMEM) and Hoechst 33342 (1 µg/mL) for 4 h (in case of U-2 OS, verapamil 10 µM was also added). Cells were washed three times with HBSS and imaged in DMEM + . Confocal microscopy images were

acquired using LEICA SP8. Gray channel ($\lambda_{ex}$ = 633 nm, $\lambda_{em}$ = 650–710 nm) corresponds to probe staining, and blue channel ($\lambda_{ex}$ = 405 nm, $\lambda_{em}$ = 415–480 nm) corresponds to Hoechst 33342 staining. Scale bar = 20 µm. This experiment was reproduced three times on different days with cells from different passages (**c**) Analysis of cell lysates (human fibroblasts (H. Fibro.), U-2 OS and HeLa CCL) using SDS-PAGE and Western blot. Predominant single fluorescent band demonstrates selective β-tubulin labeling with **6-SiR-o-C₉-CTX** (3 µM for 1 h) in living cells. This experiment was reproduced three times on different days with cells from different passages. Source data are provided as a Source data file.

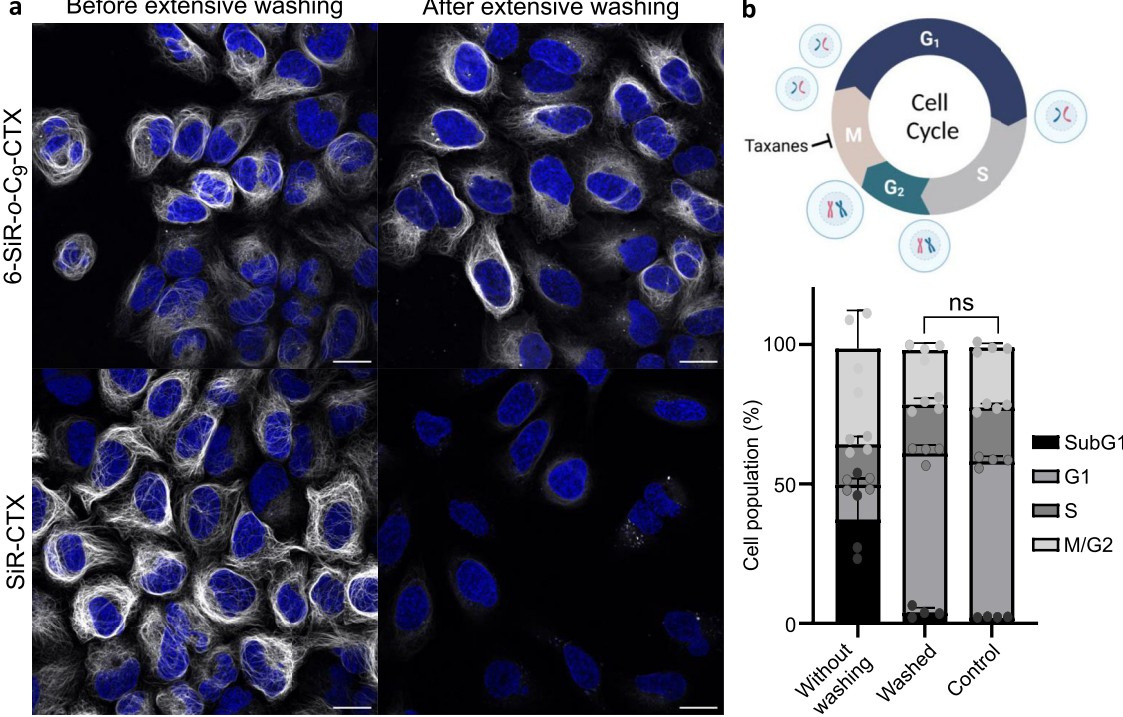

**Fig. 4 | Cleavable linker allows covalent tubulin labeling and prevents taxane cytotoxicity. a** Live-cell imaging of U-2 OS stained with probe (**6-SiR-o-C₉-CTX** – first row, **SiR-CTX** – second row, 1 µM in OptiMEM), Hoechst 33342 (1 µg/mL), and verapamil (10 µM) for 4 h before and after extensive washing (every 10 min over 2 h). Note, signal intensity is enchanted 2-folds for **6-SiR-o-C₉-CTX** (both before and after washing). Confocal microscopy images were acquired using LEICA SP8. Gray channel ($\lambda_{ex}$ = 633 nm, $\lambda_{em}$ = 650–710 nm) corresponds to probe staining and blue channel ($\lambda_{ex}$ = 405 nm, $\lambda_{em}$ = 415–480 nm) corresponds to Hoechst

33342 staining. Scale bar = 20 µm. The experiment was reproduced three times with similar results. (**b**) Cell cycle perturbation induced by **6-SiR-o-C₉-CTX** (1 µM in OptiMEM) after 24 h with or without extensive washing after 4 h incubation. Results are averages of four independent experiments (N = 4) and presented as means with standard deviations. Unpaired two-tailed t-test was performed between the 2 conditions. Cell cycle diagram created with BioRender.com. Source data are provided as a Source data file.

data showing that for this compound, most of probes are in zwitterionic form in aqueous media.

To determine the most effective probe for fluorescence microscopy, a mini-screening was performed under live-cell imaging conditions using three human cell lines: non-cancerous dermal fibroblasts and the cancerous lines HeLa CCL and U-2 OS. (Fig. 3a, b and Supplementary Figs. 14, 15). In U-2 OS cells, verapamil was added to reduce the effect of efflux pumps[37]. Probes **4** and **5** exhibited weak staining, consistent with their lower cell permeability as indicated by cytotoxicity assays. Probes **1**, **3**, and **6** effectively stained microtubules in live fibroblasts, but only faint labeling was observed in U-2 OS and HeLa CCL cells. Across all tested conditions, probe **2** (**6-SiR-o-C₉-CTX**) emerged as the most robust microtubule stain. However, occasional dot-like off-target staining was observed with **6-SiR-o-C₉-CTX** (Supplementary Fig. 16), suggesting that staining efficiency is influenced by several cell culture parameters, including metabolic state, passage number, and media age.

Next step was to check whether the bond between the protein and the dye is covalent. After staining various cell lines with **6-SiR-o-C₉-CTX**, cells were lysed (Fig. 3c), and lysates were analyzed by SDS-PAGE. *In-gel* fluorescence image revealed the presence of a strong fluorescent band at 55 kDa for **6-SiR-o-C₉-CTX** in the three cell lines tested, whereas nothing was observed for the non-covalent probe **SiR-CTX**. Western blot analysis confirmed that the fluorescent band corresponds to β-tubulin. Additional minor bands with masses exceeding 55 kDa were also detected, which we hypothesize could represent different β-tubulin isotypes and isoforms, and post-translational modifications of them[35].

The confirmation that the bond is covalent for **6-SiR-o-C₉-CTX** and not for **SiR-CTX** was further obtained by washing experiments (Fig. 4). Living U-2 OS cells were stained with either **6-SiR-o-C₉-CTX** or **SiR-CTX** at the same concentration in the presence of verapamil. After 4 h of incubation, extensive washing was performed. Images acquired before the washing process show a nice microtubule staining for both probes. After washing, the staining performed with **6-SiR-o-C₉-CTX**

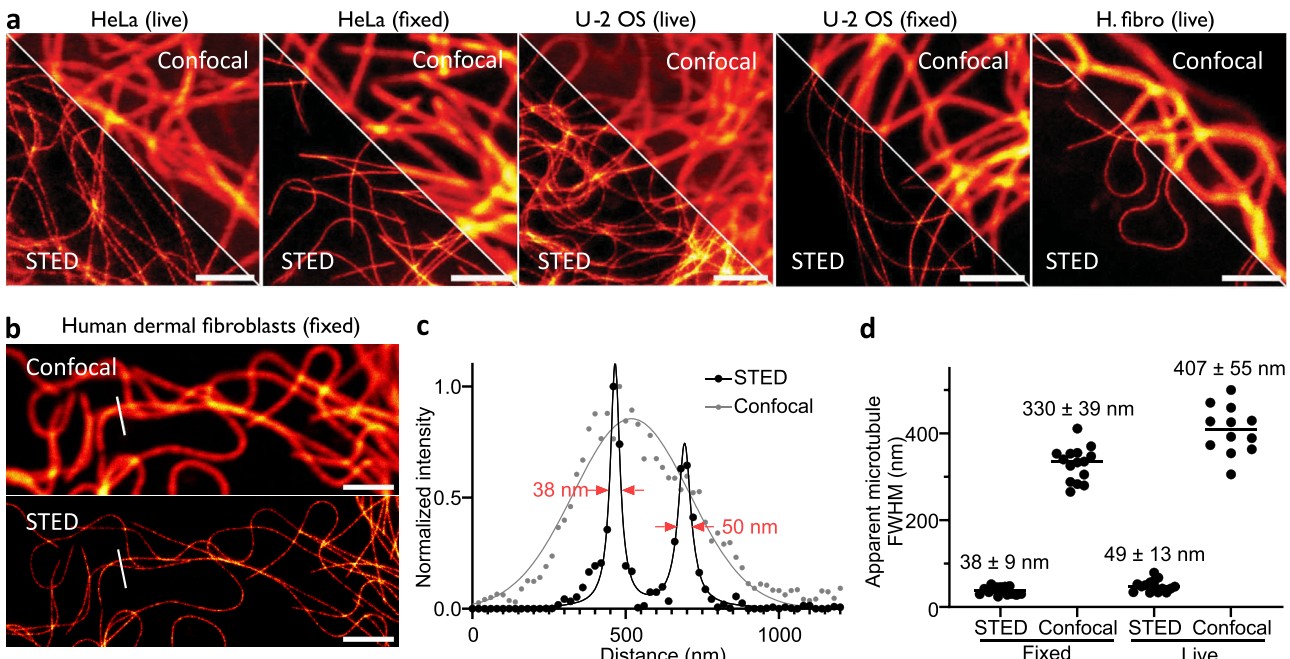

**Fig. 5 | Nanoscopy of covalently labeled tubulin in living and fixed cells.**
**a** Confocal and STED images of microtubules in HeLa, U-2 OS or human dermal fibroblast (H. fibro) cells stained with **6-SiR-o-C₉-CTX** (1 μM in OptiMEM) for 4 h. Images show optimal tubulin labeling in live and glutaraldehyde fixed cells. Images acquired with Abberior Expert Line. Scale bar = 2 μm. This experiment was reproduced three times on different days with cells from different passages (**b**) Confocal and STED images of microtubules in glutaraldehyde fixed human dermal fibroblasts stained before fixation with **6-SiR-o-C₉-CTX** (1 μM in OptiMEM) for 4 h. This experiment was reproduced three times on different days with cells from different passages (**c**) Line profile plots of fluorescence signal of the area marked in images from (**b**). **d** Quantitative analysis of apparent microtubules FWHM in human dermal fibroblasts live or fixed. The line corresponds to the mean and each dot corresponds to the result of one single measurement, from three different fields of view. N = 15 for all conditions except for live-cell confocal (N = 12). Source data are provided as a Source data file.

was still clearly visible without any significant loss of intensity, whereas for cells treated with **SiR-CTX**, the staining was not visible anymore, meaning the probe was fully washed out.

We investigated cytotoxicity of **6-SiR-o-C₉-CTX** probe 24 h after staining cells, with or without washing after 4 h incubation. In non-washed cells, a significant cytotoxic effect was observed, with approximately 40% of the SubG1 phase population. However, for extensively washed cells, no significant cytotoxicity was observed compared to the control (Fig. 4b and Supplementary Fig. 17). These results demonstrate that extensive washing effectively removes the ligand released during proximity-induced labeling, recovering cells from taxane induced phenotype and leaving labeled microtubules inside living cells.

Finally, we wanted to investigate if this new covalent probe for tubulin, **6-SiR-o-C₉-CTX**, was suitable for super-resolution microscopy and in particular stimulated emission depletion (STED) microscopy. The fluorescence of SiR can be inhibited using a 775 nm depletion laser, leading to an enhanced resolution. We stained three different cell lines (human dermal fibroblasts, U-2 OS and HeLa) with **6-SiR-o-C₉-CTX** and successfully acquired STED images of living cells or after fixation with glutaraldehyde (Fig. 5). We obtained microtubule apparent FWHM 38 ± 9 nm for fixed fibroblasts and 49 ± 13 nm living fibroblasts, which corresponds to a 8–9-fold resolution enhancement compared to confocal (Fig. 5d and Supplementary Fig. 18). In addition, the probe was photostable enough to acquire 500 s time-lapse (20 frames, Supplementary Movie 2).

## Discussion

We developed a series of covalent fluorescent probes for endogenous tubulin based on proximity induced reactivity. To do so, biocompatible benzyl sulfoniums were used as cleavable linkers.

Probes were obtained thanks to a highly convergent synthetic path, which will allow this strategy to be easily extended to other dyes and other ligands in the near future. *Ortho* isomers showed enhanced photophysical properties, reactivity, and cell permability over *meta* and *para* isomers. *Ortho*-benzyl sulfonium offers a good balance between stability and reactivity: they posses a similar labeling rate as LDAI, but can be used inside living cells because they are stable enough. We demonstrated that **6-SiR-o-C₉-CTX** can be used to covalently label endogenous tubulin in living cells after only a few hours and that this probe is suitable for STED nanoscopy. Probe is likely to be useful for labeling of microtubules in multiple species because the labeling site is mapped to a highly conserved sequence AVC̲DIPPR of β-tubulin. Introduction of cleavable linker generated tubulin probe that, after labeling releases cytotoxic taxane, which can be washed out and has a minimal impact on microtubule structure and function. Overall, this work underscores the immense potential of ligand-directed labeling and sulfonium chemistry for the covalent labeling of endogenous proteins. Our methodology offers new opportunities for advanced imaging techniques like nanoscopy and sets the stage for broader applications in cellular biology and beyond.

## Methods

### General procedure for the final step of probe synthesis

In a 1.5 mL Eppendorf, **6-SiR-COOH (1.0 eq.)** was solubilized in **dry DMSO (0.1 M)**, and **DIPEA (10 eq.)** was added. After 5 min, **TSTU (1.05 eq.)** was added. After 10 min, the formation of activated acid was controlled by LC-MS and the solution of **amine-bearing sulfonium in DMSO (1.3 eq.)**—obtained at the previous step—was added. The formation of the amide was controlled by LC-MS (it typically takes 15–20 min). Once the amide bond was formed, **TSTU (1.5 eq.)** was added, followed by **CTX-NH₃.CF₃OCO₂ (3.0 eq.)** and **DIPEA (10**

**eq.).** The formation of the final product was controlled by LC-MS. The reaction was quenched by adding **trifluoroacetic acid (50 µL)**. The crude was diluted in **ACN (1 mL)** and purified by reverse-phase HPLC (Device A (see Supplementary Information), $H_2O$ + 0.1%TFA/ACN, linear gradient 70:30–0:100, 40 mL/min). The fractions containing the product were lyophilized, and the yields were determined as followed:

The fluorescent probes were dissolved in a precise amount of DMSO-$d_6$ (600 µL) and were transferred to an NMR tube to obtain $^1H$ spectra. Afterwards the contents of the NMR tubes were transferred to an Eppendorf and were considered as stock solution. Two 1 µL samples were taken from stock solution and were diluted in Eppendorf with 59 µL of PBS containing 0.1% SDS (60-fold dilution). After 15 min, absorption of 2 µL of the diluted samples were measured on nanodrop (Nanodrop 1000, Peqlab) with 1 mm optical path. The measured absorption intensity values at the dyes absorption maxima value were averaged and concentration of the stock solution was determined according to the equation (considering that the molar absorption coefficient of the probe is comparable to that of 6-SiR-COOH $\varepsilon(\lambda_{MAX})$ = 90,000 L mol$^{-1}$ cm$^{-1}$)[29]:

$$C = \frac{Dilution\ factor * A}{\varepsilon l} \tag{1}$$

where C is the concentration of stock solution; A the sample absorption, $\varepsilon$ the extinction coefficient of the dye in PBS containing 0.1% SDS and l the path length.

Once the concentration of the stock solution was measured the amount of the obtained fluorescent conjugate could be calculated by following equation:

$$n = C * V \tag{2}$$

where C is the concentration of stock solution and V the volume of stock solution.

Finally, the yield could be determined by the classical equation:

$$\eta(\%) = \frac{n_{exp}}{n_{th}} * 100 \tag{3}$$

where $n_{exp}$ is the obtained amount of the isolated product (mol) and $n_{th}$—maximal theoretical amount of the product in the reaction (mol).

Note, the final probes of *ortho* isomers are not fully stable under the reaction conditions. Therefore, it is better to stop the reaction after 30–45 min even if the conversion is not full.

## Measurement of labeling kinetics

Tubulin from porcine brain (1 mg, Cytoskeleton, #T240, >99% Pure) was solubilized in 100 µL General Tubulin Buffer (Cytoskeleton, #BST01) supplemented with 1 mM of GTP (Thermo Fisher, #R0461) to get a 10 mg/mL solution. This solution was aliquoted, samples were snap-frozen in liquid nitrogen and stored at −80 °C prior to use.

Before the experiment, an aliquot was thawed using a water bath at room temperature. Once liquid, the sample was immediately placed on ice. In parallel, a solution of General Tubulin Buffer containing 1 mM of GTP was prepared and placed on ice.

The reaction was conducted mixing tubulin (0.5 mg/mL ~4.5 µM) and the probe (20 µM) in General Tubulin Buffer containing 1 mM of GTP at 37 °C for 48 h. Aliquots were taken at different timepoints, immediately mixed with 1/3 volume of 4x SDS sample buffer (Tris-HCl pH = 6.8: 0.2 M, SDS: 8.0% (w/v), Bromophenol Blue: 0.6 mM, glycerol: 5.4 M + 50 µL of β-mercaptoethanol per mL of solution prior to use), and boiled for 5 min at 95 °C.

The different samples were loaded on 4–15% Mini-PROTEAN® TGX™ Precast Protein Gels (Biorad, #4561086). After electrophoresis in Mini-PROTEAN® Tetra Cell using SDS-PAGE running buffer (0.25 M Tris HCl, 1.92 M glycine and 1% (w/v) sodium dodecyl sulfate (SDS) pH = 8.3), fluorescence images were recorded using Amersham Imager 600 RGB using integrated software. Quantitative data analysis was then performed using the "gel analysis" function of Fiji[7].

All experiments were performed in triplicate on different days. A gel was performed with the 48 h timepoints (3 for each compound) for all the compounds to calibrate the values accordingly. Finally, all data were calibrated with **6-SiR-*o*-C$_9$-CTX** for which the DOL was found to be 0.94 (see below).

Considering one reactant was in excess, data obtained were fitted using a single exponential (one-phase association model from Graph-Pad Prism- $Y = Y_0 + (Plateau-Y_0)*(1-exp(-K*x))$, imposing the constraint $Y_0 = 0$) to obtain the degree-of-labeling and the reaction rate for each compound.

## Sample preparation for determination of DOL (degree-of-labeling)

**6-SiR-*o*-C$_9$-CTX** (20 µM) and Tubulin (1 mg, 0.5 mg/mL ~4.5 µM, Cytoskeleton, #T240, >99% Pure) were mixed together in General Tubulin Buffer (Cytoskeleton, #BST01) supplemented with 1 mM of GTP (Thermo Fisher, #R0461) for 24 h at 37 °C. The solution was cooled at 4 °C for 1 h to depolymerize tubulin. The probe that did not react was then removed using PD MidiTrap G-25® (Cytivia). Procedure used was the one advised by the supplier, using BRB80 (80 mM PIPES pH = 6.8, 1 mM EGTA, 1 mM MgCl$_2$) as eluting buffer and conducted at 4 °C. The solution containing the protein was concentrated to the desire volume using Vivaspin® Turbo 4 (Sartorius, MWCO = 10 kDa, #VS04T02).

DOL was determined measuring $A_{280nm}$ and $A_{652nm}$ with a Nano-Drop ND-1000 spectrophotometer (Peqlab) to determine dye and protein concentrations to calculate DOL according to this formula:

$$DOL = \frac{[Dye]}{[Protein]} = \frac{\frac{A_{652nm}}{\varepsilon_{(6-SiR,\ 652nm)}}}{\frac{A_{280nm}-CF_{280nm(6-SiR)}*A_{652nm}}{\varepsilon_{(Tubulin,\ 280nm)}}} \tag{4}$$

A solution of 2% SDS (0.5 µL) was added to the solution of protein (5 µL) to ensure that the dye was fully open (three different solutions were prepared). After 30 min at room temperature, absorbance was measured at 280 nm and 652 nm. Considering $\varepsilon$(6-SiR, 652 nm) = 90,000 L mol$^{-1}$ cm$^{-1}$, $\varepsilon$(Tubulin, 280 nm) = 110,000 L mol$^{-1}$ cm$^{-1}$ and $CF_{280}$(6-SiR) = 0.147, the DOL for this sample was found to be 0.94.

## Proteomic study

**Sample preparation.** One µg of the **6-SiR-*o*-C$_9$-CTX**-labeled tubulin (as described above) was separated on 4–15% Mini-PROTEAN® TGX™ Precast Protein Gels (Biorad, #4561086) and stained with Coomassie brilliant blue. A protein band corresponding to a SiR-labeled tubulin was excised from the gel, washed, reduced with dithiothreitol (DTT), alkylated with iodoacetamide and digested with trypsin (sequencing grade, Promega) overnight. The resulting peptides were extracted, dried in a SpeedVac vacuum concentrators (Thermo Scientific) and dissolved in 2% acetonitrile/0.05% trifluoroacetic acid (v:v).

**Data acquisition.** Peptides were analyzed by electrospray ionization mass spectrometry in a Thermo Orbitrap Exploris 480 mass spectrometer coupled to an UltiMate3000 ultrahigh performance liquid chromatography system (Thermo Scientific). Chromatographic separation was performed with an in-house packed C18 reverse-

phase column (75 μm ID × 300 mm, Reprosil-Pur 120 C18-AQ, 3 μm, Dr. Maisch) using 0.1% formic acid as solvent A and 80% acetonitrile/ 0.08% formic acid as solvent B. Separating part of the HPLC method included 3 steps of linear gradients: (1) 12–42%B over 40 min, (2) 42–65%B over 27 min and (3) 65–95%B over 7.1 min. Mass spectrometer was equipped with a Nanospray Flex Ion source and controlled by Thermo Scientific Xcalibur 4.4 and Thermo Exploris 480 3.0 software. Data were acquired using an 88-min Top30 data-dependent acquisition method. One full MS scan across the 350–1600 $m/z$ range was acquired at a resolution of 120000, with an AGC target of 300% and a maximum fill time of 25 ms. Precursors with charge states 2–6 above a 1e4 intensity threshold were then sequentially selected using isolation window of 1.6 $m/z$, fragmented with nitrogen at a normalized collision energy setting of 28%, and the resulting MS2 spectra recorded at a resolution of 30,000, AGC targets of 75% and a maximum fill time of 50 ms. Dynamic exclusion of precursors was set to 22 s.

**Data processing.** Proteins and sites of SiR-labeling were identified with Thermo Proteome Discoverer 3.1.1.93 (PD) using SequestHT as a search engine. For this, Thermo raw files were searched against a database that included sequences of Sus scrofa UniProt proteome (release 24-01-2024) and common contaminants observed in MS experiments. Only trypsin-specific peptides of 6–144 amino acids were considered in the search, with maximum 3 missed cleavages. Precursor and fragment mass tolerances were set to 10 ppm and 0.02 Da, respectively. Protein N-terminal acetylation, M-oxidation, C-carbamidomethylation, as well as C/K + 573.2448 (to account for a fluorophore adduct, $C_{35}H_{35}N_3O_3Si$) were set as variable modifications. The results were filtered using the "Target Decoy PSM Validator" node in PD with default settings (FDR 0.01 and 0.05 for a "strict" and "relaxed" cut, respectively). In addition, the mass-offset search was performed with FragPipe v22.0 and MSFragger v.4.1 allowing custom mass shifts of −1.007825/0.0/114.04293/15.9949/42.0106/573.24475/79.9663 on any amino acid. The database was restricted to 13 most abundant proteins identified in a closed search.

### Information on cell lines used

Human primary dermal fibroblasts (Lonza, #CC-2511) were cultured in high-glucose DMEM (Thermo Fisher, #31053044) with 10% FBS (Thermo Fisher, #10082147) supplemented with 1 mM Sodium pyruvate (Sigma, #S8636), 1% GlutaMax (Thermo Fisher, #35050038) and 1% Penicillin-Streptomycin (Sigma, #P0781) in a humidified 5% $CO_2$ incubator at 37 °C. The cells were split every 3–4 days or at confluence.

HeLa (ATCC, CCL-2) cells were cultured in high-glucose DMEM (Thermo Fisher, #31966047) with 10% FBS (BioSELL, #S0615) supplemented with 1% Penicillin-Streptomycin (Sigma, #P0781) in a humidified 5% $CO_2$ incubator at 37 °C. The cells were split every 3–4 days or at confluence.

U-2 OS cells (ATCC, HTB-96) were cultured in McCoys 5 A medium (Thermo Fisher, #16600082) with 10% FBS (BioSELL, #S0615) supplemented with 1 mM Sodium pyruvate (Sigma, #S8636) and 1% of Penicillin-Streptomycin (Sigma #P0781) in a humidified 5% $CO_2$ incubator at 37 °C. The cells were split every 3–4 days or at confluence.

### Cell cycle analysis by imaging cytometry

The probes were dissolved in DMSO (Sigma Aldrich, #900645-4 × 2 mL) at 500–2000-fold stock concentration and added to culture media of cells at 500–2000-fold dilution accordingly. In parallel, the appropriate DMSO control samples were prepared by adding corresponding amount of DMSO volume to the separate well. HeLa cells were grown in 6-well plates (-250,000 cells per well) in presence of the fluorescent probe in variable concentrations for

24 h at 37 °C in humidified incubator with 5% $CO_2$. Cells were processed according to the NucleoCounter® NC-3000™ two-step cell cycle analysis protocol for cells attached to T-flasks, cell culture plates or micro-carriers. In particular, the 250 μL lysis solution (Solution 10, Chemometec Cat. No. 910-3010) supplemented with 10 μg/ml DAPI (Solution 12, Chemometec Cat. No. 910-3012) was used per well, incubated at 37 °C for 5 min. Then 250 μL of stabilization solution (Solution 11, Chemometec Cat. No. 910–3011) was added. Cells were counted on a NucleoCounter® NC-3000™ in NC-Slide A2™ slides (Chemometec, Cat. No. 942-0001) loaded with ~30 μL of each of the cell suspensions into the chambers of the slide. Each time, -10,000 cells in total were measured, and the obtained cell cycle histograms were analyzed with ChemoMetec NucleoView NC-3000 software, version 2.1.25.8. All experiments were repeated three times (with cells from different passages), and the results are presented as means with standard deviations.

### Cytotoxicity experiment after washing

**This protocol refers to Fig. 4b**. U-2 OS cells were seeded in 12-well plates (-300,000 cells per well) 24–48 h prior to the experiment. Cells were incubated in a humidified 5% $CO_2$ incubator at 37 °C. Two different conditions were tested: first cells were incubated for 24 h in presence of OptiMEM containing 1 μM of **Probe 2** and 10 μM of verapamil. For the second condition, cells were incubated 4 h in presence of OptiMEM containing 1 μM of **Probe 2** and 10 μM of verapamil, washed briefly 4 times with HBSS and then washed every 10 min over 2 h with DMEM+, and then further incubate with DMEM+ for 18 h. Cytotoxicity experiments were then proceed as described above. This experiment was reproduced 4 times (N = 4) on different days with cells from different passages. The results are presented as means with standard deviations.

### Western-blot

Confluent cells in a 6-well plate were incubated in the presence of OptiMEM (Thermo Fisher, #11058021) supplemented with probe (3 μM) for 1 h at 37 °C in humidified incubator with 5% $CO_2$. The media was removed, and cells were washed twice with HBSS. CelLytic™ M (300 μL per well, Sigma-Aldrich #C2978) was added, and the plate was put on a shaker for 30 min. Cell lysates were collected in Eppendorf and were centrifuged at 15,000 g and 4 °C for 20 min. Supernatants were collected and mixed with 1/3 volume of 4x SDS sample buffer (Tris-HCl pH = 6.8: 0.2 M, SDS: 8.0% (w/v), Bromophenol blue: 0.6 mM, Glycerol: 5.4 M + 50 μL of β-mercaptoethanol per mL of solution prior to use), and boiled for 5 min at 95 °C. The different samples were loaded on 4–15% Mini-PROTEAN® TGX™ Precast Protein Gels (Biorad, #4561086). After electrophoresis in Mini-PROTEAN® Tetra Cell using SDS-Page running buffer (0.25 M Tris HCl, 1.92 M Glycine and 1% (w/v) Sodium Dodecyl Sulfate (SDS) pH = 8.3), proteins were transferred from the gel to a PVDF-membrane (iBlot™ 2 Transfer Stacks, PVDF, regular size, # IB24001) using iBlot 2 Gel Transfer Device (P0 method from the supplier was used: 20 V for 1 min, 23 V for 4 min, 25 V for 2 min).

Membrane was then blocked with 1% BSA in PBS containing 0.1% of Tween 20 (Blocking buffer) overnight at 4 °C. Primary antibody (Rabbit Recombinant Monoclonal Beta-3-tubulin antibody, Abcam, #ab52623) was added (1/2000, v:v). After 1 h at room temperature, the membrane was washed three times 10 min with PBS + 0.1% Tween 20. Membrane was incubated for 1 h at room temperature in presence of a solution of secondary antibody (Donkey anti-Rabbit IgG (H + L) Highly Cross-Adsorbed Secondary Antibody, Alexa Fluor™ Plus 488, Thermo Fisher, #A32790) in blocking buffer (1/1000, v:v). After a brief washing step with PBS + 0.1% Tween 20, fluorescence images were recorded using Amersham Imager 600 RGB using integrated software.

## Sample preparation for live-cell imaging

Cells were seeded on μ-Slide 8 Well Glass Bottom dishes (Ibidi, #80827) 24–48 h prior to imaging. Cells were washed four times with HBSS (Gibco, #14025) to remove FBS and incubated with OptiMEM (Thermo Fisher, #11058021) containing 1 μM of probe in a humidified 5% $CO_2$ incubator at 37 °C. After 2 h or 4 h (depending on the experiment), the media supplemented with probe was removed, cells were washed three times with HBSS and imaged in DMEM+ [high-glucose DMEM (Thermo Fisher, #31053044) with 10% FBS (Thermo Fisher, #10082147) supplemented with 1 mM Sodium pyruvate (Sigma, #S8636), 1% GlutaMax (Thermo Fisher, #35050038) and 1% Penicillin-Streptomycin (Sigma, #P0781)].

## Sample preparation for fixed cells imaging

Protocol adapted from Gerasimaitė et al., 2021[38]. Cells were seeded on μ-Slide 8 Well Glass Bottom dishes (Ibidi, #80827) 24–48 h prior to imaging. Cells were washed 4 times with HBSS (Gibco, #14025) to remove FBS and incubated with OptiMEM (Thermo Fisher, #11058021) containing 1 μM of probe in a humidified 5% $CO_2$ incubator at 37 °C. After 2 h or 4 h (depending on the experiment), the media supplemented with probe was removed, cells were washed 4 times with 200 μL of PEMP (100 mM PIPES pH 6.8, 1 mM EGTA, 1 mM $MgCl_2$ + 4% PEG8000), permeabilized 90 s with 0.5% Triton X-100 in PEM (PEMP without PEG8000), and washed again 4 times with 200 μL of PEMP. Then, cells were incubated with 200 μL of 0.2% glutaraldehyde in PEM for 15 min, followed by 200 μL of 2 mg/mL $NaBH_4$ in PEM (dissolved immediately before use) for another 15 min. The samples were washed 4× with 200 μL of PEM and imaged in glycerol buffer (GB, 10 mM Na-$PO_4$, pH 6.8, 1 mM EGTA, 6 mM $MgCl_2$, 3.4 M glycerol).

## Live cell imaging with or without extensive washing

This protocol refers to Fig. 4a. U-2 OS were seeded on μ-Slide 8 Well Glass Bottom dishes (Ibidi, #80827) 24–48 h prior to imaging. Cells were washed 4 times with HBSS (Gibco, #14025) to remove FBS and incubated with OptiMEM (Thermo Fisher, #11058021) containing 10 μM of verapamil and 1 μM of **6-SiR-*o*-C₉-CTX** or 1 μM of **SiR-CTX** in a humidified 5% $CO_2$ incubator at 37 °C for 4 h. Cells were washed briefly 4 times with HBSS, media was replaced by DMEM+ and cells were imaged without further washing (it corresponds to the conditions "Before extensive washing"). The other condition "After extensive washing" refers to cells that were additionally washed every 10 min over 2 h with DMEM+ after the brief washing with HBSS, and then imaged in DMEM+. This experiment was reproduced three times on different days with cells from different passages with similar results.

## Confocal/STED microscopes and imaging parameters

Confocal and STED images were acquired using Abberior STED Facility Line scanning (Abberior Instruments GmbH), TCS SP8 (Leica) microscopes or Visitron Spinning disk/TIRF/SMLM system. STED images were acquired using Abberior STED Facility Line scanning (Abberior Instruments GmbH) microscopes. Imaging parameters are summarized in Supplementary Table 4.

TCS SP8 confocal microscope was controlled with Leica Application Suite X (LAS X). It is equipped with 405, 458, 476, 488, 496, 514, 561, and 633 nm excitation lasers as well as HC PL APO CS2 63x/1.40 Oil objective (Leica). Microscope has three Hybrid and two PMT detectors, which can be tuned to any detection window in the range 400–800 nm. Probes were excited at 633 nm (Laser power: 1%), and emission was collected between 650 and 710 nm.

Visitron Spinning disk/TIRF/SMLM system (Visitron Systems GmbH, Germany) is equipped with Nikon CFI Apo Lambda 60× Oil NA 1.4 objective and Prime BSI sCMOS camera (Teledyne Photometrics) with pixel size 65 nm.

Abberior STED Facility Line was controlled with Abberior instruments LigthBox (ver. 16.3.15508-w2209-win64) software. It is equipped with 488, 515, 561, 640, and 700 nm 40 MHz pulsed excitation lasers, a pulsed 775 nm 40 MHz 3 W STED laser, and an UPlanSApo 60x/1.40 Oil objective. Microscope has two APD and two MATRIX detectors, which can be tuned to any detection window in the range 400–800 nm. Pixel size was 30 nm in the xy plane was used for 2D STED images and 80 nm in the xy plane for large field of view images. Laser powers were optimized for each sample.

Abberior STED Expert Line was controlled with Imspector (ver. 16.1.7012) software. It is equipped with 561 nm and 640 nm 40 MHz pulsed excitation lasers, a pulsed 775 nm 40 MHz 3 W STED laser, and an UPlanSApo 100x/1.40 Oil objective. The following detection windows were used: for the SiR channel, 685/70 nm. Pixel size was 20 nm in the xy plane was used for 2D STED images. Laser powers were optimized for each sample.

## Visualization and modeling of the tubulin cryo-EM structures and models

Cryo-EM structures of pig tubulin complex with Taxol (PDB: 5SYF)[39] and human (PDB: 6E7C)[40] tubulin were downloaded from Protein Data Bank repository and visualized using Swiss-PdbViewer (v 4.1) or UCSF ChimeraX (v 1.6)[41]. To identify Taxol binding pocket in human protein, both structures were superimposed using iterative fit function (backbone atoms only) of the program.

## Processing and visualization of the acquired images

All acquired images were processed and visualized using Fiji[42]. Line profiles were measured using the "straight line" tool with the line width set to 3 pixels. To define apparent microtubule FWHM, line profiles were fitted with Gaussian and Lorentzian distributions for confocal and STED images, respectively.

## Statistical tests

Comparisons were performed using an unpaired t-test (two-tailed) in GraphPad Prism 10.4 (GraphPad Software, Inc., San Diego, CA, USA), and $p$-values ≤ 0.05 were considered statistically significant. All microscopy imaging experiments were repeated at least two times on different non-consecutive days (n ≥ 2). Multiple fields of view (n ≥ 3) were acquired during each imaging session, and representative images are shown in the figures.

## Reporting summary

Further information on research design is available in the Nature Portfolio Reporting Summary linked to this article.

## Data availability

The data generated in this study are provided within the paper, the Supplementary Information and the Source Data file. Mass spectrometry data available at MSV000099944 | PXD070901. Source data are provided with this paper.

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

## Acknowledgements

The authors thank the Max Planck Society for supporting this work. The authors are grateful to Dr. Vladimir Belov, Jan Seikowski, Jens Schimpfhauser and Jürgen Bienert for the NMR measurements of numerous probes and the central analytics' team (Institute for Organic and Biomolecular Chemistry, Georg-August University, Göttingen) for acquiring HRMS. The authors acknowledge the assistance of Dr. Rūta Gerasimaitė in the acquisition of spinning disk microscopy images. Figure 2d and Supplementary Fig. 10a were performed with UCSF ChimeraX, developed by the Resource for Biocomputing, Visualization, and Informatics at the University of California, San Francisco, with support from National Institutes of Health R01-GM129325 and the Office of Cyber Infrastructure and Computational Biology, National Institute of Allergy and Infectious Diseases.

## Author contributions

M.A. and G.L. conceived and planned the study. M.A., T.K., O.D., and G.L. performed the experiments. M.A., T.K., O.D., H.U., and G.L. performed the data analysis. M.A. and G.L. wrote the initial draft; all authors contributed to the final version of the manuscript.

## Funding

## Competing interests

G.L. is a co-inventor on the patent (EP2748173B1 and US9346957B2, applicant EPFL) describing SiR and its derivatives. The remaining authors declare no competing interests.
