## [Transparent Peer Review file · Nature Communications]

Biocompatible sulfonium-based covalent probes for endogenous tubulin fluorescence nanoscopy in live and fixed cells

Corresponding Author: Dr Gražvydas Lukinavičius

Version 0:

Reviewer comments:

Reviewer #1

(Remarks to the Author)

In the work, the authors reported a series of ligand-directed dyes for tubulin based on silicon-rhodamine derivative containing a fluorophore (SiR), a cleavable linker (sulfonium), and a targeting moiety (taxane). STED nanoscopy in live and fixed cells was presented using the dyes. It was noticed that similar silicon-rhodamine fluorescent probes for tubulin nanoscopy in living and fixed cells have been reported by the same group (ACS Chem. Biol. 2021, 16, 2130-2136). Although the authors explained the advantages of the ligand-directed labeling strategy over other strategies, advantages of the presented dyes in tubulin imaging over the reported dyes could be hardly observed. As the tubulin labeled with the two types of dyes exhibited similar microtubule FWHM in STED nanoscopy. Photophysical properties of the dyes are not well illustrated, and chemistry evidence for ligand-directed labeling strategy is lacked. Therefore, I would not suggest the publication of the work in Nature Communications.

Major concerns:

1. In the beginning part, it is mentioned that an alternative strategy for endogenous proteins that minimizes on the native function is aimed. But the aim may not have been realized in the presented work. As the dyes are not outstanding in tubulin imaging compared with the existing dyes.
2. Covalent labeling is highlighted. The authors did extensive washing (10 times over 2 h) to verify the covalent binding between probe 2 and the protein using SiR-CTX as reference. It seems that SiR-CTX is brighter than 6-SiR-o-C9-CTX before extensive washing. SiR-CTX might be a better choice if not the extensive washing. In other words, is extensive washing really needed before imaging? It is necessary to consider its practical application scenarios when evaluating the performance of a probe. Covalent binding may be useful for long-term tracking, where advantage of 6-SiR-o-C9-CTX over SiR-CTX might be found.
3. It well known that rhodamine dyes exist in an equilibrium between spirolactone and zwitterionic. The equilibrium is influenced by a number of factors, such as pH, solvent polarity, electronic structure of the molecule, etc. What is the driving force for the ring-opening and turn on fluorescence in Figure 1a? In addition, evidence for the mechanism of the ligand-directed labeling strategy is absent. Chemistry of the proposed dye may be studied by using proper model molecules.

Minor concerns:

1. There are some spelling errors, for example the '6-SiR-CTX' in Figure 4.
2. The references need to be checked.

Reviewer #2

(Remarks to the Author)

The manuscript by Auvray et al. reports the development of a probe for fluorescently labeling tubulin protein in nanoscopy. The probe, namely 6-SiR-o-C9-CTX, is optimized to achieve cleavable covalent binding with endogenous tubulin. The authors evaluated the cell permeability, cytotoxicity, washing to release ligand and fluorescent labeling of tubulin in both living and fixed cell.

However, the system itself does not represent much novelty as all components of this probe and their synthetic process are well-established. In addition, the characterization of the probe is not rigorous and lacks sufficiency in supporting evidence.

The authors illustrated the photophysical properties of this probe but examples of its biological applications is not robust.

Major comments:

1. Following the synthetic route described in fig. 1b, what is the purity of final product? The authors should examine and discuss the impact of the possible byproducts?
2. What is the main chirality for 6-SiR-o-C9-CTX? How the chirality affects the effectiveness of 6-SiR-o-C9-CTX?
3. Line 157 to 164 seem to be related to fig. 2c, but no indication.
4. For fig. 2c, the probe 5 and 6 showed DOL>1. The authors attributed this to multiple labeling and low selectivity. What possible labeling sites that probe 5 and 6 could bind to? Whether the other probes, including 6-SiR-o-C9-CTX, also had such labeling? These are important points to characterize the optimization.
5. For labeling sites in fig. 2d, the authors claimed all modified amino acid was cystine, indicating the selectivity of sulfonium ligand. However, the authors only add variant modifications (+573.2442) for C and K. This conclusion of labeling preference is premature unless it is performed with an open search against all amino acids. In addition, orthogonal experiment evidence, like site-specific mutation, is necessary to validate that C356 and C12 are the major modified sites of β -tubulin.
6. In fig. 4b, the authors claimed the low-toxicity of the probe based the cells in the "subG1" stage. However, this is not convincing as there is no description of how "subG1" is distinguished from "G1". In addition, this sole evidence is not rigorous enough to support the "low-toxicity".

Version 1:

Reviewer comments:

Reviewer #1

(Remarks to the Author)

All of my concerns have been addressed satisfactorily. The revisions are appropriate and have improved the quality of the paper.

I therefore recommend acceptance of the manuscript for publication.

Reviewer #2

(Remarks to the Author)

The manuscript by Auvray et al. has been revised with responses to the raised issues. However, the key weakness about novelty and biological applications remains.

Although the authors have compared the ligand-directed 6-SiR-o-C9-CTX comparing with the other dyes developed previously, there is no robust chemical evidence for its substantial strengths. For example, the authors claim the low-cytotoxicity of 6-SiR-o-C9-CTX in long incubation compared to other dyes, but did not provide convincing experiments to support it. In fact, the cytotoxic effects of the probes in this work are comparable to those in previous work by the same group (PMID: 29780462, fig. S8). Since the synthesis strategy is similar to previous work except introducing ligand-directed group which is also well established, there is not enough novelty in 6-SiR-o-C9-CTX for publication in Nature Communications.

Reviewer #1 (Remarks to the Author):

In the work, the authors reported a series of ligand-directed dyes for tubulin based on silicon-rhodamine derivative containing a fluorophore (SiR), a cleavable linker (sulfonium), and a targeting moiety (taxane). STED nanoscopy in live and fixed cells was presented using the dyes. It was noticed that similar silicon-rhodamine fluorescent probes for tubulin nanoscopy in living and fixed cells have been reported by the same group (ACS Chem. Biol. 2021, 16, 2130-2136). Although the authors explained the advantages of the ligand-directed labeling strategy over other strategies, advantages of the presented dyes in tubulin imaging over the reported dyes could be hardly observed. As the tubulin labeled with the two types of dyes exhibited similar microtubule FWHM in STED nanoscopy. Photophysical properties of the dyes are not well illustrated, and chemistry evidence for ligand-directed labeling strategy is lacked. Therefore, I would not suggest the publication of the work in Nature Communications.

We sincerely appreciate the reviewer's thorough evaluation of our work and the insightful comments. Below, we address each concern in detail and describe the revisions we have made to improve the manuscript.

Major concerns:

1. In the beginning part, it is mentioned that an alternative strategy for endogenous proteins that minimizes on the native function is aimed. But the aim may not have been realized in the presented work. As the dyes are not outstanding in tubulin imaging compared with the existing dyes.

We acknowledge the reviewer's concern regarding the novelty of our dyes compared to our previous works (ACS Chem. Biol. 2021, 16, 2130-2136) and other groups. While other studies employ silicon-rhodamine fluorophores for tubulin imaging, the current work introduces a ligand-directed covalent labeling of endogenous tubulin strategy, which distinguishes our approach from previous reversibly binding SiR-based dyes. This strategy enables selective labeling of endogenous tubulin, avoiding overexpression artifacts and preserving native cellular functions. To highlight these advantages more explicitly, we have revised the introduction and discussion sections, emphasizing the differences in labeling mechanism, cellular application, and imaging benefits.

2. Covalent labeling is highlighted. The authors did extensive washing (10 times over 2 h) to verify the covalent binding between probe 2 and the protein using SiR-CTX as reference. It seems that SiR-CTX is brighter than 6-SiR-o-C9-CTX before extensive washing. SiR-CTX might be a better choice if not the extensive washing. In other words, is extensive washing really needed before imaging? It is necessary to consider its practical application scenarios when evaluating the performance of a probe. Covalent binding may be useful for long-term tracking, where advantage of 6-SiR-o-C9-CTX over SiR-CTX might be found.

Indeed, SiR-CTX is brighter compared to the new probe, but it lacks the covalent labelling ability which is the new property and could be beneficial under certain conditions. We agree that the necessity of extensive washing should be evaluated in practical application scenarios. To clarify, extensive washing was used to rigorously demonstrate the covalent and irreversible nature of the labeling. This allows

removal of targeting moiety (taxane) which induces cytotoxicity and was already evaluated in the manuscript.

3. It well known that rhodamine dyes exist in an equilibrium between spirolactone and zwitterionic. The equilibrium is influenced by a number of factors, such as pH, solvent polarity, electronic structure of the molecule, etc. What is the driving force for the ring-opening and turn on fluorescence in Figure 1a? In addition, evidence for the mechanism of the ligand-directed labeling strategy is absent. Chemistry of the proposed dye may be studied by using proper model molecules.

We appreciate the reviewer's request for further clarification regarding the fluorescence activation mechanism. The ring-opening equilibrium of the silicon-rhodamine core is primarily influenced by the local environment, including solvent polarity and molecular interactions. The primary driving force is the difference in chemical energy of spirolactone and zwitterion. These calculations were already published in Bucevičius J. et al. (2023) Nat Commun 14, 1306. DFT calculated potential energies of zwitterion and spirolactone states against the simulated dielectric constant demonstrate that spirolactone form is more stable in dielectric environment with low dielectric constant (dioxane) and zwitterion is more stable in the environment with high value of dielectric constant (water). Free SiR probes tend to form aggregates which likely have low internal dielectric constant and this leads to high fluorogenicity. We have performed water-dioxane mix titration experiments to demonstrate this behavior of the best performing probe. We also included titration of samples containing detergent (sodium dodecyl sulphate, SDS) which effectively prevents formation of aggregates. We have modified the manuscript text appropriately and included reference.

With respect to ligand-directed labeling mechanism we have identified peptides which are labelled in the tubulin protein. The composition of the peptides includes Cys residues, which are in line with previously reported study of Cys labeling using sulfonium salts (Anal. Chem. 2022, 94, 10, 4366–4372). We have modified the manuscript text appropriately.

Minor concerns:

1. There are some spelling errors, for example the '6-SiR-CTX' in Figure 4.

We have carefully revised the manuscript and corrected all spelling errors, including the mislabeling of '6-SiR-CTX' in Figure 4.

2. The references need to be checked.

We have thoroughly reviewed and corrected the references as necessary to ensure accuracy and completeness.

Reviewer #2 (Remarks to the Author):

The manuscript by Auvray et al. reports the development of a probe for fluorescently labeling tubulin protein in nanoscopy. The probe, namely 6-SiR-o-C9-CTX, is optimized to achieve cleavable covalent binding with endogenous tubulin. The authors evaluated the cell permeability, cytotoxicity, washing to release ligand and fluorescent labeling of tubulin in both living and fixed cell.

However, the system itself does not represent much novelty as all components of this probe and their synthetic process are well-established. In addition, the characterization of the probe is not rigorous and lacks sufficiency in supporting evidence. The authors illustrated the photophysical properties of this probe but examples of its biological applications is not robust.

We would like to express our gratitude for the constructive feedback and detailed comments on our manuscript. We appreciate your time and effort in reviewing the manuscript, and we have carefully addressed the points raised. Below is our detailed response to the major comments. We have retested staining performance of the covalent probe in multiple cell lines and found that some off-targeting might be occasionally observed. This observation is mentioned in the main text together with illustrative example placed in the supplementary information. We also performed several additional probe characterization experiments listed and discussed in this point-by-point response to reviewer's letter. All these observations are discussed in the main manuscript text.

Major comments:

1. Following the synthetic route described in fig. 1b, what is the purity of final product? The authors should examine and discuss the impact of the possible byproducts?

The purity of all probes, including the best performing, is shown in Supplementary information. In particular, probe 2 has purity of >97% based on HPLC analysis. We have examined decay of the probe and found it relatively stable under physiological conditions – more than 87% remaining after 48h (Supplementary Figure 4). Indeed, free fluorophore after hydrolysis (approx. 5%) might result in increased background. Other product which shows intermolecular alkylation of taxane (approx. 3%) might also lead to increased background. However, this will not contribute to the toxicity of the probe, because none of these products should be able to bind tubulin. We have modified text accordingly.

2. What is the main chirality for 6-SiR-o-C9-CTX? How the chirality affects the effectiveness of 6-SiR-o-C9-CTX?

Indeed, we have used racemic mixture of 6-SiR-o-C9-CTX probe at sulfonium center. While the impact of chirality on binding and labeling efficiency is still under investigation, we speculate that the stereochemical configuration does influence its labeling reaction speed. On of the stereoisomers might show less reactivity compared to other.

3. Line 157 to 164 seem to be related to fig. 2c, but no indication.

Thank you for pointing this out. We have revised the manuscript text and included missing reference to Figure 2c.

4. For fig. 2c, the probe 5 and 6 showed DOL>1. The authors attributed this to multiple labeling and low selectivity. What possible labeling sites that probe 5 and 6 could bind to? Whether the other probes, including 6-SiR-o-C9-CTX, also had such labeling? These are important points to characterize the optimization.

Thank you for your insightful comment. Indeed, DOL >1 for probes 5 and 6, indicates multiple labeling sites. We assume, that one of the minor labeling sites detected in case of 6-SiR-o-C9-CTX (see supplementary figures 10 and 11) could play a role in case of probes 5 and 6. We have clarified this in the manuscript text.

5. For labeling sites in fig. 2d, the authors claimed all modified amino acid was cystine, indicating the selectivity of sulfonium ligand. However, the authors only add variant modifications (+573.2442) for C and K. This conclusion of labeling preference is premature unless it is performed with an open search against all amino acids. In addition, orthogonal experiment evidence, like site-specific mutation, is necessary to validate that C356 and C12 are the major modified sites of β -tubulin.

We fully agree with the referee and indeed checked for this by performing an MSFragger open search (looking for any mass shift with no restriction as to the amino acid identity nor position) as well as a mass-offset search (looking specifically for a dye-specific mass shift of +573.24 Da at any amino acid). Based on these results cysteine was identified as a SiR-attachment site. Although the open search suggested aspartic acid as a second much less efficient potential labeling site, this was likely due to its close position to cysteine in several SiR-labeled peptides preventing unambiguous assignment of some lower quality MS2 spectra. For the closed search (shown in our originally submitted manuscript), we have still chosen to include lysine along with cysteine as an additional modification site because of nucleophilic properties of its side chain. Regarding site-specific mutagenesis, we acknowledge its value in validating modification sites; however, given that beta-tubulin is an essential cellular protein, introducing mutations could significantly impact cell viability and function. Bacterial expression of mutant protein is even less feasible because of multiple posttranslational modifications introduced to tubulin in mammalian cells.

6. In fig. 4b, the authors claimed the low-toxicity of the probe based the cells in the "subG1" stage. However, this is not convincing as there is no description of how "subG1" is distinguished from "G1" . In addition, this sole evidence is not rigorous enough to support the "low-toxicity".

We appreciate the concern regarding the interpretation of the "subG1" stage as an indicator of low toxicity. In response, we have revised the description of the cytometry based toxicity assay. Furthermore, we have strengthened our argument by including additional evidence of low toxicity, such as cell viability assay. However, it is important to note that standard cytotoxicity tests based on plasma membrane permeation are less sensitive than our initial cell cycle assay (Figure R1). This is because DNA fragmentation (appearance of the subG1 phase) can occur before the plasma membrane becomes

permeable. The same applies to other assays that are not directly influenced by the action of taxanes. Therefore, we believe that we have employed the most relevant assays for characterizing the cytotoxicity of our probe.

Figure R1. Cytotoxicity tests based on plasma membrane permeation. U-2 OS cells were incubated for 4h with 1 μ M 6-SiR-o-C9-CTX probe and 10 μ M verapamil in Opti-MEM medium for labeling. Afterwards probe was washed off 4 times with HBSS and 10 times with DMEM + 10% FBS (each 10 min during 2 h) and cytotoxicity assay performed after additional 18 h of incubation in DMEM + 10% FBS. Control sample (“no wash”) was incubated for 24 h in Opti-MEM without probe removal.

Reviewer #1 (Remarks to the Author):

All of my concerns have been addressed satisfactorily. The revisions are appropriate and have improved the quality of the paper.

I therefore recommend acceptance of the manuscript for publication.

We sincerely appreciate the reviewer's evaluation and time dedicated to review our work.

Reviewer #2 (Remarks to the Author):

The manuscript by Auvray et al. has been revised with responses to the raised issues. However, the key weakness about novelty and biological applications remains.

We sincerely thank the reviewer for the careful evaluation of our work and value the opinion expressed.

Although the authors have compared the ligand-directed 6-SiR-o-C9-CTX comparing with the other dyes developed previously, there is no robust chemical evidence for its substantial strengths. For example, the authors claim the low-cytotoxicity of 6-SiR-o-C9-CTX in long incubation compared to other dyes, but did not provide convincing experiments to support it. In fact, the cytotoxic effects of the probes in this work are comparable to those in previous work by the same group (PMID: 29780462, fig. S8). Since the synthesis strategy is similar to previous work except introducing ligand-directed group which is also well established, there is not enough novelty in 6-SiR-o-C9-CTX for publication in Nature Communications.

We appreciate the reviewer's comment regarding the novelty of our dyes in comparison to our previous work (Chem. Sci., 2018, 9, 3324–3334). The key innovation of the current study lies in the ligand-directed covalent labeling of endogenous tubulin using sulfonium salt cleavable linker, which fundamentally differentiates our approach from the previously reported reversibly binding SiR-based dyes. This covalent labeling strategy enables selective and stable modification of endogenous tubulin, thereby avoiding overexpression artifacts and maintaining native cellular functions. As a result, the probe exhibits reduced cytotoxicity after washing (Figure 4) and preserves fluorescence signals over extended periods, even after cell fixation (Figure 5d and Supplementary Figure 18).

While ligand-directed labeling strategies have been reported previously, none employ a cleavable linker containing a sulfonium salt. We propose that this electrophile offers enhanced biocompatibility compared to earlier systems, as structurally related sulfonium species naturally occur in cells in form of S-adenosyl-L-methionine (AdoMet).